# A New Look at the Swing Contract: From Linear Programming to Particle Swarm Optimization

**Tapio Behrndt [1,*] and Ren-Raw Chen [2]**

1   Gasum Oy, Revontulenpuisto 2C, 02100 Helsinki, Finland
2   Gabelli School Business, Fordham University, 45 Columbus Avenue, New York, NY 10019, USA; rchen@fordham.edu
*   Correspondence: tapio.behrndt@gmail.com

**Abstract:** As the energy market has grown in importance in recent decades, researchers have paid increasing attention to swing option contracts. Early studies evaluated the swing contract as if it were a financial derivative contract, by ignoring its storage constraints. Aided by recent advances in artificial intelligence (AI) and machine learning (ML) technologies, recent studies were able to incorporate storage limitations. We make two discoveries in this paper. First, we contribute to the literature by proposing an AI methodology—particle swarm optimization (PSO)—for the evaluation of the swing contract. Compared to the other ML methodologies in the literature, PSO has an advantage by expanding to include more features. Secondly, we study the relative impact of the price process (exogenously given) that underlies the swing contract and the storage constraints that affect a quantity decision process (endogenously decided), and discover that the latter has a much greater impact than the former, indicating the limitation of the earlier literature that focused only on price dynamics.

**Keywords:** swing option; linear programming; dynamic programming; artificial intelligence; particle swarm optimization

## 1. Introduction

One of the distinguishing features of the energy markets is the difficulty of storing the underlying commodities at a low cost over a long time period. At one end of the spectrum we have crude oil, which can be stored relatively easily, but at the other end of the spectrum we have electricity, which essentially cannot be stored efficiently (even though some electricity can be stored in batteries, the storage capacity is still very limited). Somewhere in the middle of these two extremes we have natural gas, which can be stored in highly specialized storage facilities underground, or in its liquefied form, which requires natural gas to be cooled down to $-162\,°C$. In particular, the storage of natural gas is a highly complex technical problem and traditional gas consumers will not be able to maintain their own natural gas storage from which to draw on when demand spikes.

As a direct consequence, many of the commonly traded derivative contracts in the natural gas market are designed to allow flexibility in the delivery of the underlying commodity, both in terms of timing and volume. These types of derivative contracts are commonly referred to as "swing contracts".

As an example, consider a natural gas consumer who uses gas in order to produce heat. Usually such a consumer will buy a base load of gas on a daily basis in order to generate sufficient heat, however, on a particularly cold day it might be necessary for this consumer to buy an extra quantity of gas in order to produce more heat and meet a potential spike in demand. So, the consumer requires some flexibility in the purchase volume on a daily basis that might deviate from the actual daily forecasts. Buying this excess gas on the spot market might be extremely costly as there might be a run on gas by consumers on cold

winter days and spot prices might spike. In 13–17 February of 2021, a strong winter storm Uri devastated North America (United States, northern Mexico, and parts of Canada). In Texas, it had caused an unprecedented electricity shortage. Due to its "free-market" energy policies, the State of Texas allows the utility companies to transfer the actual electricity costs to consumers.[1] A distinguishing feature of these demand spikes is that they usually revert quickly to their mean, and this type of flexibility in delivery of extra volumes is only required on a few days throughout the winter.

There are a few straightforward, though sub-optimal, ways to hedge against such an event. For example, the consumer could hedge the risk by buying a daily strip of European options that would allow the consumer to purchase a fixed quantity of gas at a fixed price every single day. Of course, this would be a very costly and inefficient way to hedge this risk, since in reality only a few of those daily options would be exercised due to the mean reverting behavior of the gas price.

A swing contract allows its holder to purchase a flexible quantity of the underlying commodity at a fixed price and a fixed future date subject to local (e.g., daily) and global (e.g., all-time) storage constraints. This flexibility is an attractive choice, for example, for natural gas consumers and matches their risk profile better than the previously mentioned hedging alternatives. For example, a swing contract could allow its holder to buy a certain amount of gas on a daily basis throughout the winter (daily decision within daily constraints), subject to a total consumption limit. In addition, on each day, the buyer of the swing contract can also consume any quantity of natural gas (but up to the storage limit). As a result, it is a derivative contract that must be balanced between price and quantity in order to maximize the profit (or minimize the cost).

Swing contracts are commonly traded over-the-counter and allow their holders to purchase a flexible quantity of the underlying asset at fixed exercise dates, subject to local and global constraints. For example, a power producer relying on wind to generate power might have entered into an agreement to deliver a certain amount to the power supply on a monthly basis. In case the wind does not produce sufficient amounts of energy, they would be able to hedge their delivery risk by purchasing a swing contract on power. In the same way, a gas consumer might require volume flexibility throughout the winter in order to balance their delivery risk.

While the methodologies used to evaluate the swing contract are quite standard (e.g., dynamic programming (DP), also known as lattice), the implementations are extremely expensive due to the complex structure of the contract. The purpose of this paper is two-fold. First, we show that, with a slight modification, the problem can be solved via linear programming (LP), which is extremely fast. The error is shown to be small. Secondly, we propose the use of an artificial intelligence (AI) method, known as the particle swarm optimization (PSO), as an alternative to the existing expensive methods. PSO (or any AI method) in general is not fast, yet it is robust to high dimensional problems, and as a result, perfectly suitable for complex contracts, such as the swing option.

## 2. The Swing Contract and the History of Pricing Models

Kohrs et al. (2019), who provide an excellent review of the literature and the evolution of the commodity derivative market, explain the swing contract very well. In their words, swing contracts " . . . incorporate flexibility-of-delivery options, known as 'swing' or 'take-or-pay' options, which allow the holder to repeatedly exercise the right to receive greater or smaller quantities of energy subject to local daily and global periodic constraints". As a result, a swing contract is a complex/exotic commodity derivative contract that must be balanced between price and quantity in order to maximize the profit (or minimize the cost).

There are a large number of variations of the swing contract and the literature was not clear and varies to a large degree on how to differentiate various contractual terms in the swing contract. Often swing contracts are mixed with what is known as a storage contract. Strictly speaking, a swing contract is an option contract between the supplier of the commodity (natural gas) and the buyer of the commodity (utility firm). However,

the utility firm has a storage concern which interferes with how the utility firm negotiates the swing contracts with its suppliers. This raises massive confusion for those who are interested in studying the swing contract. At the same time, the literature has made various simplifying assumptions in order to make the valuation tractable. Here, we follow the existing literature to combine the two contracts with some of those simplifying assumptions.

It is clear that swing contracts are path-dependent American-style derivatives. As mentioned earlier, there is no closed-form solution to this pricing problem and complex numerical algorithms are a must to calculate the value of the contract.

Various numerical techniques were tested to evaluate swing contracts. As part of the general derivative family, swing contracts (regardless of the variations) can be naturally evaluated by the lattice method (also known as the dynamic programming method). Jaillet et al. (2004) were the first to adopt a comprehensive multi-variate lattice (known as "forest of trees") to evaluate the swing contract. However, as expected, due to the complexities of the swing contract, their lattice is costly to implement with a large number of nodes. As a common substitute for the lattice, the Monte-Carlo-Least-Square (MCLS) method, first proposed by Longstaff and Schwartz (2001), is popular in pricing American-style options, as recently used by Boogert and de Jong (2011), among various other earlier researchers.

Besides the standard valuation methods, such as lattice and MCLS, more advanced algorithms were proposed in the literature. Bonnans et al. (2012) adopt a stochastic control approach, which generate more accurate swing contract prices.[2] Given that American-style derivatives are an optimal stopping time problem, Carmona and Touzi (2008) adopt the theory of the Snell envelope to evaluate the swing contract, which is a multi-stopping time problem. Kluge (2006) employs an integration method and compares the results with Monte Carlo bounds, provided by Meinshausen and Hambly (2004). Bardou et al. (2009) and Bonnans et al. (2012) adopt the optimal quantization method [3] and compare it with the least-square Monte Carlo simulation method (LSMC) and conclude that the optimal quantization method is far more superior to the LSMC method (at least in an unconstrained case).

As a natural continuation of the option valuation, hedging implications are widely discussed in the literature. For example, Keppo (2002) and Warin (2012) discuss hedging of the swing contract and Hambly et al. (2009) study the impact of price spikes in the energy market. Recognizing that swing contracts are not the same as typical financial options, and hence standard hedging does not apply, Pflug and Broussev (2009) use the game theory to model the behavior of the seller.

In addition to the horserace of various numerical algorithms, the literature also includes extensions to the simplest model proposed by Jaillet et al. (2004). Jaillet et al. (2004); Boogert and de Jong (2011); Bonnans et al. (2012), among others, assume the underlying energy price is driven by multiple factors. Wahab et al. (2010) assume the underlying energy price is governed by a regime switching process. Safarov and Atkinson (2017) assume the underlying energy price is governed by a complex Levy process. Thompson et al. (2009) use the real option technique, which assumes a very flexible underlying asset price process.

Kohrs et al. (2019) provide an excellent review of the literature above and interested readers are referred to their original paper for the history and review. An earlier review can be found in Løland and Lindqvist (2008).

The common deficiency of all of the above models is that quantity limitations are not considered. In other words, the above models treat this energy derivative as a financial derivative in which the quantity in transactions is completely elastic. As a result, the Texas situation discussed in the Introduction can be incorporated nowhere in their models.

However, incorporating quantity in the valuation substantially deviates from the traditional option pricing methodologies. This is because the classical pricing models assume the quantity to be completely elastic (or known as infinite supply). In other words, quantity has no impact on price. Once quantity has an impact on price, the classical models fail, in the sense that no reasonable stochastic process can describe the actual price

movements in the market (as in Texas). Not to mention that the decisions involved in setting the quantity (subject to limitations) can themselves impact on the price of the swing option. The discussion of how quantity can impact on the price of energy can be traced back to Holland (2007), and yet only until now has the technology caught up with the reality of the market.

It is natural then for researchers to look for answers in various artificial intelligence and machine learning areas. Daluiso et al. (2020) argue that the swing option is effectively a stochastic control problem with a set of actions and, as a consequence, use reinforcement learning for the valuation. Curin et al. (2021) amalgamate the reinforcement learning with MCLS. Malyscheff and Trafalis (2017) integrate the stochastic vector machine with MCLS.

As in the more recent, machine learning-related literature, we regard the swing contract as a combination of an option (i.e., the holder of the contract has the right to buy the underlying commodity at a strike price) and a decision-making process (i.e., the holder can decide the amount of the underlying commodity to buy and sell). Doing it this way, not only can we clarify a number of confusing issues in the literature, but we can easily extend our model to evaluate exceptionally complex swing contracts.[4]

Let $0 = t_0 < t_1 < \cdots < t_n = T$ be the times where the swing contract can be exercised. Usually (and so assumed from now on), these times are daily. In addition, let $q_{min} < q_{max}$ and $Q_{min} < Q_{max}$ be the daily quantity limits and global quantity limits allowed in the swing contract. $S(t)$ represents the (spot) price of the underlying asset and $K$ is the strike price.

A swing contract is a derivative contract on $S(t)$ and allows its holder to buy a daily quantity $q_i$ at each exercise date $t_i$ where $i = 1, \cdots, n$ at the strike price $K$ subject to the following:

$$q_{min} \leq q_i \leq q_{max}$$
$$Q_{min} \leq \sum_{i=1}^{n} q_i \leq Q_{max} \tag{1}$$

We should note that in some swing contracts those bounds are not hard bounds but subject to penalties, i.e., the holder of the contract can buy above the local/global limit, however, they will pay a penalty, which is usually a fixed percentage of the price of the underlier (i.e., something such as the Henry Hub price + 20%). For simplicity, the literature treats those penalties as infinite, and we assume the same.

To add to the literature, we propose the use of a PSO (particle swarm optimization) algorithm to evaluate the swing contract. There are two advantages of using PSO for the valuation of the swing contract. First, PSO can be easily combined with Monte Carlo simulations and hence gain computational efficiency. Second, PSO can more easily combine price (i.e., stochastic process exogenously given) and quantity information (i.e., decision-making process endogenously decided) in the valuation. Furthermore, PSO allows easy expansions to more complex contracts or allows for more sources of randomness (as suggested by Jaillet et al. (2004) and Boogert and de Jong (2011) to adopt multiple factors).

In addition to the novelty of adopting PSO in valuing the swing contract, which is the first time in the literature, we also provide some insights toward how price and quantity interact and their relative contribution to the price of the swing contract. In particular, we discover that the buy/sell decision plays the dominant role in the price of the swing option, while the price process does not. As a result, the problem can be very easily resolved just by using linear programming.[5] As discussed toward the end of the paper, for the price process to have an impact, it must generate enough higher moments. In other words, the price impacts are in the situations where risk-neutral pricing fails. This is an important empirical question to answer. We conclude that the theory papers in the literature (including this paper) that are predominantly based upon continuous-time martingale processes for the natural gas will not be able to generate a substantial price impact on the swing contract. In such a case, a simple linear programming algorithm can produce satisfactory results for the swing contract.

### 3. The Valuation

As documented in the literature, there are a number of moving parts in the swing contract: (1) stochastic price process; (2) random interest rate environment; (3) lack of liquidity in the market (and hence a utility dependent valuation is necessary); and (4) quantity optimization. We summarize all of the moving parts and write the valuation equation as follows:

$$C(t) = \max_{q_i} \sum_{i=1}^{n} \mathbb{E}_t \left[ \Lambda(t, T_i) \{ -1_{q_i>0} q_i \min\{\Phi(T_i, T_i + \Delta), K\} - 1_{q_i<0} q_i S(T_i) \} \right] \quad (2)$$

subject to Equation (1):

$$q_{\min} \leq q_i \leq q_{\max}$$

$$Q_{\min} \leq \sum_{i=1}^{n} q_i \leq Q_{\max}$$

where $1_{\{\cdot\}}$ is the indicator function; $T_i$'s are exercise dates; $\Phi(t, T_i)$ is the futures price purchased at time $t$ and delivered at time $T_i$; $S(T_i)$ is the spot price; $\Phi(T_i, T_i + \Delta)$ is the $T_i$ futures price settled at $T_i + \Delta$ ($\Delta$ is usually a day); $K$ is the strike price; $q_i$ is the quantity sold (–) or purchased (+) at time $T_i$; $\Lambda(t, T_i)$ is known as the pricing kernel; and $\mathbb{E}_t$ is the conditional expectation under the physical measure.

In Equation (2), we note that the pricing kernel $\Lambda(t, T_i)$ carries the risk premium (of the representative agent). We also note that the pricing kernel is also known as the marginal rate of substitution between periods.[6]

Equation (2) states that in a swing option contract, the owner can choose to buy and sell repeatedly at specific dates during the contract period $\underline{T} = < T_1, \cdots, T_n >$. When selling, the owner will receive the spot price. When buying, the owner will pay the lower of the strike price and the futures price (not the spot price). The reason why it is the futures price is because the delivery of the natural gas to the owner of the swing option is delayed (by $\Delta$, which is usually a day). The quantity $q_i$ at time $T_i$ is either positive (determined by the indicator function) upon purchase or negative at sale. The owner will calculate the optimal quantity to buy or sell at each exercise time $T_i$ in order to maximize the total profit over the contract period.

We note that Equation (2) is not a straightforward option pricing problem. It includes a decision-making process. The contract allows its owner to decide how much quantity to buy (by exercising the option at the strike price $K$) and sell (in the cash market at the spot price $S(T_i)$), subject to the daily and all-time limits ($q_{\max}$, $q_{\min}$ and $Q_{\max}$ and $Q_{\min}$, respectively). An optimized decision is made at each exercise date $T_i$ based upon the best knowledge given at that time.

Hence, while Equation (2) is easy to write down, its implementation is not. First of all, pricing kernel, price and quantity are all random (and could be correlated). The optimal decision of quantity $q_i$ is made on the fly at each time $T_i$. Hence, a natural way to solve the problem is dynamic programming (i.e., a lattice), which optimizes backwards along the price lattice. This is not an easy implementation.

As a result, apparently, Equation (2) is not exactly implemented in the literature. Equation (2) can be greatly simplified as follows (subject to Equation (1)):

$$C(t) = \max_{q_i} \sum_{i=1}^{n} P(t, T_i) \hat{\mathbb{E}}_t \left[ \{ 1_{q_i>0} q_i \min\{\Phi(T_i, T_i + \Delta), K\} + 1_{q_i<0} q_i S(T_i) \} \right] \quad (3)$$

where $P(t, T_i)$ is the risk-free discount factor (to replace the pricing kernel $\Lambda(t, T_i)$) and $\hat{\mathbb{E}}_t$ is the conditional expectation under the risk-neutral measure (to replace the conditional expectation under the physical measure $\mathbb{E}_t$).

From Equation (2) to Equation (3) is the well-known change of measure. The change of measure (from physical to risk-neutral) requires either (1) continuous trading (and in a frictionless market), or (2) a representative utility function. Given that energy commodities are unlike financial assets and cannot be transacted easily and in small portions, a common simplification adopted is to assume that the representative agent is risk-neutral, i.e., $\hat{\mathbb{E}}_t =$

$\mathbb{E}_t$. As a consequence of this assumption, the pricing kernel is equal to the risk-free discount factor.

Apparently, even Equation (3) is highly complex and expensive to implement (see Jaillet et al. 2004). Not only is the optimal decision on quantity and buy/sell at each period a result of current and future prices, but it is also a result of future buy/sell and quantity decisions.[7]

### 3.1. Deterministic Quantities

As long as the quantity decision is not made dynamically, the valuation becomes dramatically easier. The assumption made in this sub-section is that the quantity decision is made at the current time $t$, and hence there is no uncertainty of quantity. When the quantity of each period is fixed, then Equation (3) becomes extremely easy to solve, as demonstrated below:

$$
\begin{aligned}
C(t, \underline{T}; \underline{q}) &= \max_{q_i} \sum_{i=1}^{n} P(t, T_i) 1_{q_i>0} q_i \hat{\mathbb{E}}_t[\min\{\Phi(T_i, T_i+\Delta), K\}] + 1_{q_i<0} q_i \hat{\mathbb{E}}_t[S(T_i)] \\
&= \max_{q_i} \sum_{i=1}^{n} P(t, T_i) \{1_{q_i>0} q_i[\Phi(t, T_i+\Delta)(1-\Pi_i^+) - K\Pi_i^-] + 1_{q_i<0} q_i \Phi(t, T_i)\}
\end{aligned}
\tag{4}
$$

Finally, $\Pi_i^+$ and $\Pi_i^-$ are two probabilities similar to those in the Black–Scholes model, indicating the likelihood of the option being in-the-money (but under different probability measures), which are derived below:[8]

In Equation (4), we note that the risk-neutral expectations of future spot price and futures price are both futures' prices. If the futures price follows the Black model (1976):

$$
\frac{d\Phi(t, T_i)}{\Phi(t, T_i)} = \sigma dW(t)
\tag{5}
$$

then the two probabilities are normal:

$$
\Pi_i^{\pm} = N\left(\frac{\ln \Phi(t, T_i) - \ln K}{\sigma\sqrt{T_i - t}} \pm \frac{1}{2}\sigma^2(T_i - t)\right)
\tag{6}
$$

where $N(\cdot)$ is the standard normal probability. Note that Equation (4) does not have a closed form solution (although the price option does). Equation (4) needs to be solved for a series of quantities (positive is buy and negative is sell/consume). Given that the objective function is linear (both $\Phi(t, T_i+1\text{d})\Pi_i^+ - K\Pi_i^-$ and $\Phi(t, T_i)$ are known), the problem can be solved via linear programming:

$$
\max_{q_i} \sum_{i=1}^{n} P(t, T_i) \{1_{q_i>0} q_i a_i + 1_{q_i<0} q_i b_i)\}
\tag{7}
$$

subject to Equation (1):

$$
q_{\min} < q_i < q_{\max}
$$

$$
Q_{\min} < \sum_{i=1}^{n} q_i < Q_{\max}
$$

Then,

$$
a_i = \Phi(t, T_i+1\text{d})\Pi_i^+ - K\Pi_i^-
$$

$$
b_i = \Phi(t, T_i)
$$

This is a linear programming problem that can be easily solved in Excel using the `Solver`. An example will be provided in the next section (Section 5) where both the linear programming and dynamic programming are presented and compared.

### 3.2. Stochastic Quantities Yet Uncorrelated with Prices

When the quantities are random (i.e., determined at each date) but independent of prices, then the problem can also be greatly simplified. Equation (3) can be rewritten as follows:

$$
\begin{aligned}
C(t) \quad &= \max_{q_i} \sum_{i=1}^{n} P(t, T_i) \times \\
&\quad \left\{ \hat{\mathbb{E}}_t \left[ \{1_{q_i>0} q_i\} \right] \hat{\mathbb{E}}_t [\min\{\Phi(T_i, T_i + \Delta), K\}] + \hat{\mathbb{E}}_t \left[ 1_{q_i<0} q_i \right] \hat{\mathbb{E}}_t [S(T_i)] \right\} \\
&= \max_{q_i} \sum_{i=1}^{n} P(t, T_i) \times \\
&\quad \left\{ \hat{\mathbb{E}}_t \left[ \{1_{q_i>0} q_i\} \right] \left\{ \Phi(t, T_i + \Delta)(1 - \Pi_1) - K \Pi_2 \right\} + \hat{\mathbb{E}}_t \left[ 1_{q_i<0} q_i \right] \Phi(t, T_i) \right\}
\end{aligned}
\tag{8}
$$

With the prices already computed, we only need to optimize the quantities. As a result, a dynamic programming approach can be used. In the dynamic programming (lattice), we can identify an optimal path.

The state space can be set up in the following manner, as shown in Figure 1:

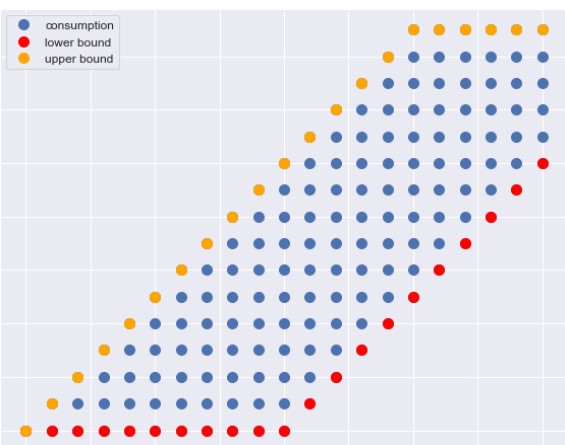

**Figure 1.** Consumption, Lower Bound and Upper bound.

At the beginning (current time), the holder of the swing contract has no inventory and hence can only buy. Over time, the holder can only buy up to the maximum capacity limit $Q_{\max}$—labeled by the yellow dots. Once the limit is reached, the holder can no longer buy, but only sell or take no action. The red dots represent the lower limit $Q_{\min}$ and in between is the daily most allowable capacity. As a result, the holder must start buying since the daily allowable capacity is reached. Note that the diagram is just a demonstration. The actual state space needs to be constructed according to the actual contract.

Now the quantities to buy or sell $q_i$ can be decided at each node. Note that this is a one-dimensional lattice that is quite different from the lattice forest by Jaillet et al. (2004). This one-dimensional lattice is extremely fast to solve.

An explicit example will be provided later (Section 5) and will be compared to linear programming. We discover that the results are identical. In other words, the optimal path in the lattice is the same as the result of linear programming where the quantities are totally deterministic. In other words, the quantities, whether random or not, do not impact the solution, as long as they are independent of the price.

### 3.3. Full Model

When the quantities are determined at each future exercise time $T_i$ and are correlated with the underlying price process, then the numerical algorithm to solve Equation (3) can instantly become more complex, as shown in Jaillet et al. (2004) and the others reviewed in the previous section.

To evaluate the full model, we propose an alternative artificial intelligence (AI) method: particle swarm optimization (PSO) to "intelligently" search for the optimal quantity decision process. AI methods are in general less preferred than parametric numerical methods, such as lattice or PDE. However, in the case of the swing option, none of the parametric methods can provide an accurate solution cheaply. As a result, we contend that PSO has a merit in reaching a solution reasonably fast and cheaply and should be considered as a practical alternative.

We also compare the PSO result of the full model (Equation (3)) with the results of the linear programming (limited model, Equation (4)) and find small differences. This brings confidence in using the limited model as a fast approximation.

## 4. Particle Swarm Optimization (PSO)

In theory, swarm intelligence is effective for optimization problems in a high-dimensional space. PSO is such an application. The original version of PSO was first proposed by Eberhart and Kennedy (1995) who modified the behavioral model of swarm into an objective-seeking algorithm. Similar to Reynold's, their model "artificializes" the group behavior of a flock of birds seeking food. Via bird-to-bird chirping (peer-to-peer communication), all of the birds fly to the loudest sound of chirping. Subsequently, Shi and Eberhart (1998) improved the model by adding an inertia term (symbolized as $w$ later as we introduce the model) and it has become the standard PSO algorithm used today. To set a proper value of the inertia term is to seek the balance between *exploitation* and *exploration*. A larger value of the inertia term gives more weight to exploration (as the bird is more likely to fly on its own), and a smaller value of the inertia term gives more weight to exploitation (as the bird intends more to fly toward other birds).[9]

One can compare PSO to the grid search. A grid search can find the global optimum and yet it takes an exploding amount of time to reach such a solution, especially in a high-dimensional space. PSO can be regarded as a "smart grid search" where each particle performs a "stupid search" and yet, by communicating with other particles and by having a large number of such particles, we can reach the global optimum quickly.

Imagine we need to measure the deepest place of a lake whose bottom has an uneven surface. A two-dimensional grid search can easily find the global minimum. An alternative would be PSO. Imagine we have a number of "fish" (particles) who swim in the lake. At each time step, all of the fish will measure the depth of the lake underneath them. Each fish is communicating with all of the other fish to decide whose depth is the deepest (minimum). All of the fish now remember the minimum and then they swim for another time step. At each time step, they update the global minimum so far. If we let these fish swim randomly for enough time, we will reach the global minimum.

In the case of the lake, we may find the grid search to be more accurate and time-effective. However, in an $n$-dimensional lake, grid searches are becoming ineffective, but the same number of fish may just do the same job with the same amount of time as in the two-dimensional lake.

Currently there were a limited number of applications of PSO in finance, mostly in the portfolio selection. Chen et al. (2021) uses it for the first time in the literature to locate the exercise boundary of American-style derivatives (specifically, put option, option on min/max, and Asian option).

The PSO algorithm can be formally defined as follows. For $i = 1, \cdots, n$ particles and each particle is a vector of $j = 1, \cdots, m$ dimensions, we have:

$$\begin{cases} \vec{v}_{i,j}(t+1) = w(t)\vec{v}_{i,j}(t) + r_1 c_1 (\vec{p}_{i,j}(t) - \vec{x}_i(t)) + r_2 c_2 (\vec{g}(t) - \vec{x}_{i,j}(t)) \\ \vec{x}_{i,j}(t+1) = \vec{x}_{i,j}(t) + \vec{v}_{i,j}(t+1) \end{cases} \tag{9}$$

where $\vec{v}_{i,j}(t)$ is velocity of the $i$th particle in the $j$th dimension at time $t$; $\vec{x}_{i,j}(t)$ is position of the $i$th particle in the $j$th dimension at time $t$; $w(t)$ is a "weight" (less than 1), which



decides how the current velocity will be carried over to the next period (and usually it is set as $w(t) = \alpha w(t-1)$ and $\alpha < 1$ to introduce diminishing velocity);[10] and finally $r_1, r_2 \sim u(0,1)$ follow a uniform distribution.

In the swarm literature, $w(t)\vec{v}_i(t)$ is called inertia; $r_1 c_1(\vec{p}_i(t) - \vec{x}_i(t))$ is called the cognitive component; and $r_2 c_2(\vec{g}(t) - \vec{x}_i(t))$ is called the social component. Coefficients $c_1$ and $c_2$ are known as acceleration coefficients.

At each position there is "cost function" $f(\cdot)$ (sometimes called distance function) at which a "cost" (or penalty) is computed. This cost function is the objective function to be minimized (or maximized).

The global best at any given time is either the maximum or minimum value of the objective function generated by all of the particles at the time:

$$\vec{g}(t) = \min_i \left\{ f(\vec{p}_i(t)) \right\} \tag{10}$$

and the personal best at the time is:

$$\vec{p}_i(t) = \min_t \left\{ f(\vec{x}_i(t)) \right\} \tag{11}$$

and $f(\cdot) : \mathbb{R}^n \to \mathbb{R}$ is the "fitness function". The usual fitness function is

$$f(\vec{x}_i(t)) = \|x_i - \underline{\chi}\| = \sum_{j=1}^{J} (x_{ij} - \chi_j)^2 \tag{12}$$

where $\underline{\chi} = < \chi_1, \cdots, \chi_J >$ is a coordinate in a $J$-dimensional space.

As we can see, the algorithm (at least the standard one presented here) of PSO is quite different from that of a generic swarm by Reynolds (1987). Yet they both share the same behavioral pattern of a natural swarm. In other words, (1) both PSO and the generic swarm are based upon peer-to-peer communication in order to achieve the objective, and (2) the particles in both PSO and the generic swarm are identical (such as birds or ants) and each particle follows its neighbor particles. The difference is just how each particle weighs its neighbors. In PSO, each particle only cares about the global best discovered by its neighbors, and in the generic swarm each neighbor's position is important. We provide an example to demonstrate the mechanical details of PSO in Appendix A.

## 5. A Demonstration

In this section, we first demonstrate a limited model (deterministic quantity case) of Equation (4). In this case, we can solve the problem via easy linear programming (LP). We also demonstrate how dynamic programming (DP) and particle swarm optimization (PSO) can lead to the same solution. This implies that the price dynamics are relatively less important in determining the price of the swing option contract and the quantity decision dominates. To see why, we note that Equation (2) is an expectation on the physical measure where risk preference matters. Yet, in the demonstration below, the Black model, which assumes (log) normality, is adopted. Under the (log) normal distribution, risk-neutral measure can be easily derived since the (log) normal distribution has only two moments. In a more complex distribution where higher moments matter, then the risk-neutral measure is not easily achievable (unless the representative agent has quadratic utility function (see Merton (1973) for a full discussion)). Then Equation (2) cannot be simplified and the only solution to the problem must be an AI/ML one, such as PSO. This is an empirical question and we leave it to future research.

Then, we will solve the full model of Equation (3) or (8). The full model can only be solved via DP or PSO. We demonstrate that PSO is highly efficient in solving complex option problem, such as the swing option. In the literature (reviewed earlier), reinforcement learning (Daluiso et al. 2020) and stochastic vector machine (Malyscheff and Trafalis 2017) were used to solve the problem. While in the finance literature, comparisons of various models are useful in identifying the best model, we do not provide a horserace of the various

AI/ML tools in this paper. This is because such comparisons are highly case dependent and algorithm-dependent (i.e., how the code is written). Unless such comparisons are performed in a wide collection of transactions (i.e., an empirical study), differences are usually not meaningful. Hence, this is different from the large volume of horserace papers in finance. We argue that PSO is easy to visualize and then better to be introduced to the industry. Judging from the methodology, SVM and PSO are very similar and reinforcement learning is more tedious to program than PSO.[11]

*5.1. Limited Model*

To make the problem simple, so that we have a known solution, we adopt a simple example, as follows. The prices given in this demonstration are either $5 or $10. The action the investor can take is either to buy or sell one unit at a time (i.e., $q_{min} = -1$ and $q_{max} = 1$); or he can take no action (i.e., $q_i = -1, 0, 1$). There is a storage limit of five units (i.e., $Q_{max} = 5$). Finally, the contract holder cannot short sell (minimum capacity is 0 or $Q_{min} = 0$). The tenor of the contract is assumed to be 22 days.

Intuitively, one could guess the solution to be buying low (at $5) and selling high (at $10). Such a profit is $50, which is the maximum. We shall see that the LP solution is consistent with that intuition. We also discover that the DP solution is the same as the LP solution, although the investor is allowed to dynamically optimize his action (and the inventory as a result).

5.1.1. Linear Programming

Table 1 is an example where the prices are known for the next 22 exercise dates (imagine this is a daily schedule, so roughly a month). They are given in column 2 of Table 1.

**Table 1.** A Hypothetical Example of a Swing Contract.

| Day | Prices | Action |
|-----|--------|--------|
| 1 | 5 | 1 |
| 2 | 5 | 1 |
| 3 | 10 | −1 |
| 4 | 10 | −1 |
| 5 | 5 | 1 |
| 6 | 5 | 1 |
| 7 | 10 | −1 |
| 8 | 10 | −1 |
| 9 | 5 | 1 |
| 10 | 5 | 1 |
| 11 | 10 | −1 |
| 12 | 10 | −1 |
| 13 | 5 | 1 |
| 14 | 5 | 1 |
| 15 | 10 | −1 |
| 16 | 10 | −1 |
| 17 | 5 | 1 |
| 18 | 5 | 1 |
| 19 | 10 | −1 |
| 20 | 10 | −1 |
| 21 | 5 | 1 |
| 22 | 5 | −1 |

The maximum profit is $50. In Table 1, "1" means "buy", and the investor spends the price to acquire either 5 or 10 units of goods; and "−1" means "sell", and the investor makes money by selling the goods at the given price. For example, at day 1, the investor has no inventory to sell so s/he must buy (or no action) at $5. Hence, the day-1 profit is −$5. At day 2, the investor can either buy more, or sell the one unit s/he just bought. In

the solution, he would choose to buy one more unit. The reason is that the price is low so s/he keeps buying and then sells at a higher price at $10, which is exactly what s/he does on day 3. S/he sells one unit at $10 at day 3 and another unit at $10 at day 4. It is clear that the investor will always buy at a low price and sell at a high price, provided that he cannot sell short, nor can he accumulate inventory over the given capacity.

Hence, "1" means "buy", "−1" means "sell", and "0" means no action. The investor makes money by selling, and pays money when buying.

LP can easily solve the problem (e.g., Excel `Solver`[12]) and the solution is presented in column 3 of Table 1. The maximum profit is $50.

### 5.1.2. Dynamic Programming

A dynamic programming approach is also implemented. The state space for DP is given in Table 2. The numbers in the table represent the quantities held by the investor at a specific time and state. Given the maximum capacity of five units, the state space can be only up to five.

**Table 2.** State Space of Dynamic Programming.

| Day | 0 | 1 | 2 | 3 | 4 | 5 | 6 | 7 | 8 | 9 | 10 | 11 | 12 | 13 | 14 | 15 | 16 | 17 | 18 | 19 | 20 | 21 | 22 |
|---|---|---|---|---|---|---|---|---|---|---|---|---|---|---|---|---|---|---|---|---|---|---|---|
|  |  |  |  |  |  | 5 | 5 | 5 | 5 | 5 | 5 | 5 | 5 | 5 | 5 | 5 | 5 | 5 |  |  |  |  |  |
|  |  |  |  |  | 4 | 4 | 4 | 4 | 4 | 4 | 4 | 4 | 4 | 4 | 4 | 4 | 4 | 4 | 4 |  |  |  |  |
|  |  |  |  | 3 | 3 | 3 | 3 | 3 | 3 | 3 | 3 | 3 | 3 | 3 | 3 | 3 | 3 | 3 | 3 | 3 |  |  |  |
|  |  |  | 2 | 2 | 2 | 2 | 2 | 2 | 2 | 2 | 2 | 2 | 2 | 2 | 2 | 2 | 2 | 2 | 2 | 2 | 2 |  |  |
|  |  | 1 | 1 | 1 | 1 | 1 | 1 | 1 | 1 | 1 | 1 | 1 | 1 | 1 | 1 | 1 | 1 | 1 | 1 | 1 | 1 | 1 |  |
|  | 0 | 0 | 0 | 0 | 0 | 0 | 0 | 0 | 0 | 0 | 0 | 0 | 0 | 0 | 0 | 0 | 0 | 0 | 0 | 0 | 0 | 0 | 0 |

This table presents the inventory level over time (22 days). The capacity limit is five units which can be reached in 5 days (permitted action every day is to buy or sell one unit). At the end of the month, it is optimal that there is no inventory left (all sold), which implies that day 18 is the last day that the inventory level can be five. The actions permitted on each day is then clear. On day 1, the investor can either do nothing (0)—resulting in no inventory on day 1; or buy (1)—resulting in one unit of inventory on day 1. S/he cannot sell since that would result in negative inventory, which is not allowed. On each of the consecutive days, the investor can buy, no action, or sell. The optimal action is given in Table 1.

In an extreme case, the investor can buy from day 1 for 5 consecutive days and reach the storage capacity. In another extreme case, the investor holds on to a maximum number of units of five till the last allowable day (which is 5 days till maturity), in that it is optimal for the investor to sell all of the goods at maturity. Hence, the state space steps down starting day 17 and reaches 0 on day 22.

The result (i.e., payoff) of the DP is given in Table 3. Table 3 is similar to Table 2, except that the quantity numbers are replaced by payoffs. In DP, we need to move backwards from maturity.

**Table 3.** Payoff of Dynamic Programming.

| Day | 0 | 1 | 2 | 3 | 4 | 5 | 6 | 7 | 8 | 9 | 10 | 11 | 12 | 13 | 14 | 15 | 16 | 17 | 18 | 19 | 20 | 21 | 22 |
|---|---|---|---|---|---|---|---|---|---|---|---|---|---|---|---|---|---|---|---|---|---|---|---|
| Price |  | 5 | 5 | 10 | 10 | 5 | 5 | 10 | 10 | 5 | 5 | 10 | 10 | 5 | 5 | 10 | 10 | 5 | 5 | 10 | 10 | 5 | 5 |
|  |  |  |  |  |  | 65 | 65 | 60 | 55 | 55 | 55 | 50 | 45 | 45 | 45 | 40 | 35 | 35 |  |  |  |  |  |
|  |  |  |  |  | 60 | 60 | 60 | 55 | 50 | 50 | 50 | 45 | 40 | 40 | 40 | 35 | 30 | 30 | 30 |  |  |  |  |
|  |  |  |  | 60 | 55 | 55 | 55 | 50 | 45 | 45 | 45 | 40 | 35 | 35 | 35 | 30 | 25 | 25 | 25 | 20 |  |  |  |
|  |  |  | 60 | 55 | 50 | 50 | 50 | 45 | 40 | 40 | 40 | 35 | 30 | 30 | 30 | 25 | 20 | 20 | 20 | 15 | 10 |  |  |
|  |  | 55 | 50 | 50 | 45 | 45 | 40 | 40 | 35 | 35 | 30 | 30 | 25 | 25 | 20 | 20 | 15 | 15 | 10 | 10 | 5 | 5 |  |
|  | 50 | 45 | 40 | 40 | 40 | 35 | 30 | 30 | 30 | 25 | 20 | 20 | 20 | 15 | 10 | 10 | 10 | 5 | 0 | 0 | 0 | 0 | 0 |

From this result, we can see the maximum profit is $50 on day 0. The backward induction process of the dynamic programming is explained in the text. It is quite lengthy and so we will not repeat it here.

The optimal path is presented via the shaded nodes. We can see that on day 0 there are two choices of action: buy or no action. If buy, the $5 is spent and if no action, then $0 is spent. The best result is to buy rather than no action because the payoff of buy is $55 − $5 = $50 and greater than the payoff of no action, which is $45 + $0 = $45 (note that $50 and $45 are payoffs on day 1 from backward induction). At state 1 on day 1, three choices of action are available: buy again, no action, and sell. The payoff of each decision is $60 − $5 = $55, $50 + $0 = $50, and $40 + $5 = $45, respectively. Hence, clearly the optimal decision is to buy again. At state 2 on day 2, the price goes up to $10. There are three choices of action: buy, no action, and sell. The resulting payoffs are $60 − $10 = $50, $55 + $0 = $55, and $50 + $10 = $60, respectively, so clearly the optimal decision is to sell. This process continues and the shaded boxes indicate the optimal action path.

At maturity, the payoff is $0 because there is no inventory at hand. On day 21, if the investor has no inventory at hand (i.e., state 0 in Table 2), he will have nothing to sell and hence make no money. Certainly he will not buy, because he has no chance to sell it and will suffer a $5 loss (Or −$5). Choosing the larger of the two, at state 0, the investor will only take no action, and hence the resulting payoff is $0. On day 21, there is another possible state—state 1. At state 1, the investor has one unit in inventory and can sell it. This is the only permissible action, and selling the unit will generate $5 payoff.

| Day | 21 | 22 |
|---|---|---|
| state 1 | 5 | |
| state 0 | 0 | 0 |

Moving backwards to day 20, there are three possible states: 0, 1, and 2 (reflecting the inventory). At state 0, the investor has two possible actions: buy or no action. The buy action will incur a cost of $5 and will move on to state 1 on day 21, which has a payoff of $5. Spending $5 now and receiving $5 the next day will result in $0. Alternatively, the investor can take no action. Under this action, the investor will move on to state 0 on day 21, which pays $0. As a result, no action on day 20 also yields $0, which is the same as buying. Taking the larger of the two same results, the payoff at state 0 on day 20 is $0.

At state 1 on day 20, only sell and no action are allowed (since there is no state 2 on day 21). If the investor takes no action, then he will end up with $5 the next day. If the investor sells, then he will receive $5 from selling and $0 from the payoff (state 0) of the next day. Both results are equal, and hence we know that the payoff at state 1 on day 20 is $5.

At state 2 on day 20, the only permissible action is to sell. This yields $5 from selling and $5 from the next day (state 1), and so together is $10, as follows:

| Day | 20 | 21 | 22 |
|---|---|---|---|
| state 2 | 10 | | |
| state 1 | 5 | 5 | |
| state 0 | 0 | 0 | 0 |

Repeating the process, we can derive all of the payoffs in the lattice, as shown in Table 3. As we can see, the maximum payoff on day 0 is $50. We can trace the best decision path by following the combined result of action and the next period payoff. The optimal path is highlighted in the shaded cells. The explanation of the optimal path is offered alongside the table to easily compare the numbers.

Quite amazingly, the result of this DP algorithm is the same as the result of LP. Random quantities (i.e., deciding what action to take on the fly) make no difference in the results of LP (i.e., deciding the action at the beginning), once the prices are deterministic. In the next sub-section, we implement PSO to demonstrate in this simple example how in detail we can construct a PSO algorithm to solve the LP problem.

### 5.1.3. PSO

A PSO algorithm can also easily find the solution, although much more slowly than LP or DP. In the PSO, we first randomly assign a location to each particle (i.e., fish/bee/ant). The location is a vector of all 22 of the actions over the period. The following is the VBA code:

- `If Rnd > (2/3) Then fish(ifsh, idim) = 1`
- `If Rnd < (2/3) And u > (1/3) Then fish(ifsh, idim) = 0`
- `If Rnd < (1/3) Then fish(ifsh, idim) = −1`

where `ifsh` indicates the *i*th fish and `idim` (for dimension) represents the *i*th action and hence `fish(ifsh, idim)` is the action of *i*th fish on the *i*th day. As we can see, the location is randomly assigned 1 (sell), 0 (no action), and −1 (buy). The three actions are chosen equally from a uniform random number `Rnd` (which is an VBA function to call upon a uniform random number).

We also need to choose the initial velocity:

- `velo(ifsh, idim) = Rnd`

After the initialization, particles (fish) start to move around. These subsequent movements are decided by new velocities.

The velocity in the following iterations is determined by combining three amounts: from itself; from its personal best; and from the global best, as indicated by the right-hand-side of Equation (9). However, the right-hand-side of Equation (9) is not directly the velocity to be used to update either the next velocity or the position, but needs to be translated into integers. This is because our actions are only allowed to be 1, 0, and −1, which is not the general case of a PSO. Then this velocity is added to the previous location to arrive at the next location of each particle (fish), as Equation (9) describes.

A sample run is given in Table 4. The number of particles used in this example is 100. As we can see it converges rather quickly (in iteration 6).[13]

**Table 4.** PSO.

| Day | 1 | 2 | 3 | 4 | 5 | 6 | 7 | 8 | 9 | 10 | 11 | 12 | 13 | 14 | 15 | 16 | 17 | 18 | 19 | 20 | 21 | 22 |
|---|---|---|---|---|---|---|---|---|---|---|---|---|---|---|---|---|---|---|---|---|---|---|
| Price | 5 | 5 | 10 | 10 | 5 | 5 | 10 | 10 | 5 | 5 | 10 | 10 | 5 | 5 | 10 | 10 | 5 | 5 | 10 | 10 | 5 | 5 |
| Profit | | | | | | | | | | | actions | | | | | | | | | | | |
| 15 | 0 | 1 | 0 | −1 | 1 | 1 | −1 | −1 | 1 | 0 | −1 | 0 | 0 | 0 | 0 | 0 | 0 | 0 | 0 | 1 | 0 | −1 |
| 15 | 0 | 1 | 0 | −1 | 1 | 1 | −1 | −1 | 1 | 0 | −1 | 0 | 0 | 0 | 0 | 0 | 0 | 0 | 0 | 1 | 0 | −1 |
| 30 | 1 | 1 | −1 | −1 | 1 | 1 | −1 | −1 | 1 | 1 | −1 | −1 | 0 | 0 | 0 | 0 | 1 | 0 | 0 | −1 | 1 | 0 |
| 40 | 1 | 1 | −1 | −1 | 1 | 1 | −1 | −1 | 1 | 1 | −1 | −1 | 0 | 1 | −1 | 0 | 1 | 0 | 0 | −1 | 1 | −1 |
| 40 | 1 | 1 | −1 | −1 | 1 | 1 | −1 | −1 | 1 | 1 | −1 | −1 | 0 | 1 | −1 | 0 | 1 | 0 | 0 | −1 | 1 | −1 |
| 45 | 1 | 1 | −1 | −1 | 1 | 1 | −1 | −1 | 1 | 1 | −1 | −1 | 1 | 1 | −1 | 0 | 1 | 0 | −1 | −1 | 1 | −1 |
| 50 | 1 | 1 | −1 | −1 | 1 | 1 | −1 | −1 | 1 | 1 | −1 | −1 | 1 | 1 | −1 | −1 | 1 | 1 | −1 | −1 | 1 | −1 |
| 50 | 1 | 1 | −1 | −1 | 1 | 1 | −1 | −1 | 1 | 1 | −1 | −1 | 1 | 1 | −1 | −1 | 1 | 1 | −1 | −1 | 1 | −1 |
| 50 | 1 | 1 | −1 | −1 | 1 | 1 | −1 | −1 | 1 | 1 | −1 | −1 | 1 | 1 | −1 | −1 | 1 | 1 | −1 | −1 | 1 | −1 |
| 50 | 1 | 1 | −1 | −1 | 1 | 1 | −1 | −1 | 1 | 1 | −1 | −1 | 1 | 1 | −1 | −1 | 1 | 1 | −1 | −1 | 1 | −1 |

particles = 100.

Table 4 presents the days and daily prices in the top two rows, below which are the iterations and the result of each iteration. The profit is given on the left and the daily actions taken are given afterwards. The daily actions are the "gbest" among all of the particles (fish/bees/ants)—that is the result of the best fish.

As we can see, at the first iteration, the best fish does not offer a good solution. The actions suggested only result in $15 profit. This is expected and the coordinates (i.e., actions) are randomly chosen. In the next iteration, there is no improvement (daily actions are same), indicating the best fish in the initial iteration remains the best fish. In iteration 3, another fish takes over and the profit increases to $30. In about six iterations, PSO reaches the optimal solution.

### 5.2. Full Model

In this sub-section, we consider the general case where both the prices and quantities are random and furthermore correlated. DP in this case is very complex. This is because once the prices are random, actions (quantities) are dependent on the prices given at each time and node. The lattice must be first built for the prices. Then at each time and node, the decision of an optimal action is made. This is the same as lattice on lattice (known as "lattice forest").

Complex lattices always run into computational issues—either memory or speed. Usually they are not able to provide accurate results, due to computation limitations. To overcome this dimensionality curse problem, we employ PSO as an alternative to DP or PDE. In PSO, we apply, for each particle (fish), a vector of actions (one for each month for 12 months, i.e., $< x_1, \cdots, x_{12} >$) are applied on the $N$ simulation paths of prices. Then, via iterations, the best action is determined and the option value computed. Detailed discussions of using PSO to evaluate the expectation, such as Equation (3), can be found in Chen et al. (2021).

This is identical to the lattice algorithm, in that all of the fish swim through all of the simulation paths. Hence, in an $M \times N$ ($M$ fish and $N$ price paths) universe, the best action is decided and option price calculated.

To carry out this exercise, we employ real data. The data we use in this demonstration are obtained from the CME website[14], which contains the prices of all of the futures contracts on Henry Hub natural gas (per million BTU, or British thermal unit) on a given day. The data are presented (for the next 12 months) in Table 5. The spot data are obtained from the EIA (U.S. Energy Information Administration).[15]

**Table 5.** CME Futures Prices (per million BTU) 1/13/2021.

| Month | Last | Change | Prior Settle | Open | High | Low | Volume |
|---|---|---|---|---|---|---|---|
| 02/01/21 | 2.743 | −0.01 | 2.753 | 2.737 | 2.826 | 2.708 | 157769 |
| 03/01/21 | 2.703 | −0.004 | 2.707 | 2.69 | 2.773 | 2.671 | 83168 |
| 04/01/21 | 2.7 | 0.001 | 2.699 | 2.679 | 2.758 | 2.676 | 58202 |
| 05/01/21 | 2.718 | 0.002 | 2.716 | 2.705 | 2.772 | 2.7 | 28261 |
| 06/01/21 | 2.783 | 0.011 | 2.772 | 2.773 | 2.826 | 2.76 | 17321 |
| 07/01/21 | 2.854 | 0.013 | 2.841 | 2.833 | 2.893 | 2.831 | 17359 |
| 08/01/21 | 2.87 | 0.014 | 2.856 | 2.86 | 2.91 | 2.849 | 7340 |
| 09/01/21 | 2.858 | 0.016 | 2.842 | 2.845 | 2.894 | 2.835 | 5438 |
| 10/01/21 | 2.877 | 0.014 | 2.863 | 2.854 | 2.914 | 2.852 | 23347 |
| 11/01/21 | 2.931 | 0.015 | 2.916 | 2.92 | 2.965 | 2.909 | 3597 |
| 12/01/21 | 3.053 | 0.014 | 3.039 | 3.025 | 3.083 | 3.025 | 3609 |
| 01/01/22 | 3.136 | 0.013 | 3.123 | 3.111 | 3.166 | 3.111 | 11949 |

Spot price 2.82. https://www.eia.gov/dnav/ng/hist/rngwhhdm.htm. (accessed on 13 January 2021).

We simulate spot prices of natural gas using the Black–Scholes model:

$$\frac{\mathrm{d}S(t)}{S(t)} = r(t)\mathrm{d}t + \sigma\mathrm{d}W(t) \tag{13}$$

where $r_t$ is the risk-free rate (assumed to be a deterministic process); $\sigma$ is volatility (assumed to be constant); and $W_t$ is the risk-neutral Brownian motion. The futures prices can be computed as:

$$\begin{aligned} \Phi(t, u) &= \mathbb{E}_t[S(u)] \\ &= S(t) \exp\left(\int_t^u r(w)\mathrm{d}w\right) \\ &= S(t)M(t, u) \end{aligned} \tag{14}$$

As a result, we can compute the money market account as:

$$M(t, u) = \frac{\Phi(t, u)}{S(t)} \tag{15}$$

This is used in simulation:

$$\ln S(t + \Delta t) = \ln S(t) + M(t, t + \Delta t) + \sigma \sqrt{\Delta t} \varepsilon(t) \tag{16}$$

where $\varepsilon(t)$ is the standard normal random number sampled at time $t$.

While these are stochastic prices, the PSO used here is similar to the one in the limited model. Each particle (fish) contains a vector 22 action values—one per day for the whole 22-day period. Each fish in the PSO "swims" (i.e., trying various values of the vector) and compares with other fish in terms of the value of the objective function—profit from the actions. Given that our example is to buy or sell at futures prices, the value of the objective function is just the inner product of the action vector and futures price curve.

The steps of PSO are given as follows. Let the action vector be $\underline{x} = \{x_1, \cdots, x_{22}\}$ for the 22-day period. Each fish has such a vector and competes with other fish for the best vector (i.e., "gbest") as follows:

1.  set a specific decision process (i.e., vector of buys/sells);
2.  simulate standard normal variables (22 periods by 1000 paths);
3.  calculate price of the swing option by Equation (16) (22 periods by 1000 paths);
4.  calculate the expected payoff: $\frac{1}{1000} \sum_{i=1}^{22} \sum_{j=1}^{1000} x_i S(t_i)$;
5.  find the best fish (with the highest value);
6.  revise the decision process;
7.  repeat steps #2~5 until it converges

The above process is similar to the American option valuation, introduced in Chen et al. (2021). A brief sketch of their process is introduced in Appendix A.

The result of PSO is given in Table 6. The market futures prices are given in row 2 and the calibrated money market account (MM) is given in row 3. These MM account values are computed using (15) and used in simulations to guarantee the risk-neutral expectations converge to the market futures prices in row 2. The volatility is set at 50% and the number of simulation paths is 1000. The number of particles is 400.

**Table 6.** PSO Result.

| Day | 1 | 2 | 3 | 4 | 5 | 6 | 7 | 8 | 9 | 10 | 11 | 12 |
|---|---|---|---|---|---|---|---|---|---|---|---|---|
| Price | 2.743 | 2.703 | 2.7 | 2.718 | 2.783 | 2.854 | 2.87 | 2.858 | 2.877 | 2.931 | 3.053 | 3.136 |
| MM | −0.0277 | −0.0147 | −0.0011 | 0.00664 | 0.02363 | 0.02519 | 0.00559 | −0.0042 | 0.00663 | 0.0186 | 0.04078 | 0.02682 |

| iter | gbest | Day = 1 | 2 | 3 | 4 | 5 | 6 | 7 | 8 | 9 | 10 | 11 | 12 |
|---|---|---|---|---|---|---|---|---|---|---|---|---|---|
| 1 | −0.771 | 0 | 0 | 1 | 0 | 1 | 0 | 1 | 0 | 0 | −1 | −1 | −1 |
| 2 | −0.79016 | 0 | 0 | 1 | 0 | 1 | 0 | 1 | 1 | −1 | −1 | −1 | −1 |
| 3 | −1.0969 | 0 | 1 | 1 | 1 | 1 | 1 | 0 | −1 | −1 | −1 | −1 | −1 |
| 4 | −1.22089 | 1 | 1 | 1 | 1 | 1 | 0 | −1 | 0 | −1 | −1 | −1 | −1 |
| 5 | −1.22089 | 1 | 1 | 1 | 1 | 1 | 0 | −1 | 0 | −1 | −1 | −1 | −1 |
| 6 | −1.22089 | 1 | 1 | 1 | 1 | 1 | 0 | −1 | 0 | −1 | −1 | −1 | −1 |
| 7 | −1.22089 | 1 | 1 | 1 | 1 | 1 | 0 | −1 | 0 | −1 | −1 | −1 | −1 |
| 8 | −1.22089 | 1 | 1 | 1 | 1 | 1 | 0 | −1 | 0 | −1 | −1 | −1 | −1 |

The parameters are given as follows:

| | |
|---|---|
| S(t) | 2.82 |
| volatility | 0.5 |
| npaths | 1000 |
| nfish | 400 |

Note: In the above table, the top part contains the price information as before. MM is the marginal rate of substitution representing the expected (see Equation (15)). The bottom part is the list of PSO results in each iteration. As we can see, in four iterations, PSO has reached the optimal solution (hence, iterations six–eight are redundant). The actions are

given in columns 3~14 where 1 = buy; 0 = no action; and −1 = sell. The optimal value (gbest) is $1.22.

In a few steps,[16] PSO converges to $1.2209 and the action is given accordingly on the right. We note that this result is the same as the LP result, which is $1.22, as presented in Table 7. The slight difference could be due to a Monte Carlo error.

**Table 7.** LP Result.

| Day | 1 | 2 | 3 | 4 | 5 | 6 | 7 | 8 | 9 | 10 | 11 | 12 |
|---|---|---|---|---|---|---|---|---|---|---|---|---|
| Price | 2.743 | 2.703 | 2.7 | 2.718 | 2.783 | 2.854 | 2.87 | 2.858 | 2.877 | 2.931 | 3.053 | 3.136 |
| MM | −0.0277 | −0.0147 | −0.0011 | 0.00664 | 0.02363 | 0.02519 | 0.00559 | −0.0042 | 0.00663 | 0.0186 | 0.04078 | 0.02682 |
| 1.22 | 1 | 1 | 1 | 1 | 1 | 0 | −1 | 0 | −1 | −1 | −1 | −1 |

Note: In the above table, the top part contains the price information as before. MM is the marginal rate of substitution representing the expected (see Equation (15)). The linear programming result is the same as the PSO result in Table 6. The maximum profit is $1.22 at the bottom of column 1. The actions are given in columns 2~13.

The results of this example indicate that correlated random prices and quantities do not add value to the option price. This result could be due to the fact that the permissible actions are only limited (i.e., buy or sell only one unit). As we expand the decision space (and hence the state space), the impact is larger. Still, we note that a larger state space creates computational challenges for DP and PDE, but not for PSO.

## 6. Conclusions

The swing option contract was evaluated in the literature for nearly three decades. While it was recognized as a storage contract (as early as Holland 2007), earlier valuation models continued to use price process(es) as the only pricing dynamics, mainly due to the computational limitations. Recently, thanks to the growing trading activities in the energy markets and the technological advances in the artificial intelligence (AI) and machine learning (ML) areas, a renewed interest in valuing the swing contract has emerged and storage constraints are explicitly considered.

Joining this growing interest of valuing the swing contract using AI/ML models, we propose the use of a particle swarm optimization (PSO) methodology. Compared to the other ML methodologies in the literature, PSO has an advantage in expanding to include more features. In particular, PSO is more streamlined with Monte Carlo simulations, and hence can gain computational efficiency. Moreover, PSO can more easily combine price (i.e., the stochastic process exogenously given) and quantity information (i.e., the decision-making process endogenously decided) in the valuation. Lastly, PSO allows easy expansions to more complex contracts, or allows for more sources of randomness (as suggested by Jaillet et al. 2004 and Boogert and de Jong 2011 to adopt multiple factors).

In addition to the novelty of adopting PSO in valuing the swing contract, which is the first time that this was completed in the literature, we also provide some insights toward how price and quantity interact and their relative contribution to the price of the swing contract. We discover that the buy/sell decision plays the dominant role in the price of the swing option, while the price process does not. As a result, the problem can be very easily resolved just using linear programming. For the price process to have an impact, it must generate enough higher moments. In other words, the price impacts are in the situations where risk-neutral pricing fails. This is an important empirical question to answer. We conclude that the theory papers in the literature (including this paper) that are based mostly upon continuous-time, martingale processes for the natural gas will not be able to generate substantial price impact on the swing contract. In such a case, a simple linear programming algorithm can produce satisfactory results for the swing contract.

One can use PSO to model more complex variations of the swing contract with few difficulties. Many swing contracts involve multiple assets (different gas prices at different terminals). To solve this problem, a non-linear optimization (yet still integer and most

likely binary in the shipping variable, as demonstrated earlier) may be effective. We leave this for future research.

**Author Contributions:** Conceptualization, T.B. and R.-R.C.; methodology, T.B. and R.-R.C.; software, T.B. and R.-R.C.; validation, T.B. and R.-R.C.; formal analysis, T.B. and R.-R.C.; investigation, T.B. and R.-R.C.; resources, T.B. and R.-R.C.; data curation, T.B. and R.-R.C.; writing—original draft preparation, T.B. and R.-R.C.; writing—review and editing, T.B. and R.-R.C.; visualization, T.B. and R.-R.C.; project administration, T.B. and R.-R.C. All authors have read and agreed to the published version of the manuscript.

**Funding:** This research received no funding.

**Institutional Review Board Statement:** This research requires no ethical approval.

**Informed Consent Statement:** This research involves no humans.

**Data Availability Statement:** This research uses no data.

**Acknowledgments:** We thank Xiaoyu Deng for her capable assistance and benefit from Thomas Chu for his valuable suggestions. We also are grateful to two anonymous referees for their valuable suggestions.

**Conflicts of Interest:** The authors have no conflict of interest.

## Appendix A

*(a) A PSO Demonstration*

In this appendix, we demonstrate the mechanics of PSO used in this paper. We adopt a simple sphere as the objective function, so we know the answer of the solution:

$$f(x_1, x_2) = (x_1 - 0.5)^2 + (x_2 - 0.5)^2 \tag{A1}$$

Hence the solution is:

$$x_1 = 0.5$$
$$x_2 = 0.5$$

To conserve space and easy reading for the readers, we explain the details of the PSO implementation by Table A1 so the readers can refer to the table more easily.

Note: The parameters of PSO in Equation (9):

$$\begin{cases} \vec{v}_{i,j}(t+1) = w(t)\vec{v}_{i,j}(t) + r_1 c_1(\vec{p}_{i,j}(t) - \vec{x}_i(t)) + r_2 c_2(\vec{g}(t) - \vec{x}_{i,j}(t)) \\ \vec{x}_{i,j}(t+1) = \vec{x}_{i,j}(t) + \vec{v}_{i,j}(t+1) \end{cases}$$

are given below. Except for rare cases, the results of PSO are usually quite insensitive to these parameters. As noted in the text, $c_1$ and $c_2$ amplify exploitation and exploration, respectively. We set them as equal to not bias the model. $w$ is usually less than one in order to encourage convenience:

| | |
|---|---|
| c1 | 1 |
| c2 | 1 |
| w | 0.9 |

We demonstrate PSO using two iterations. Each fish carries a coordinate in the x–y plane. These values are shaded in gray in the table. The day-0 values are randomly created between 0 and 1. The values of the objective function of the five fish are ( shaded in yellow ) $f(x, y) = (x - 0.5)^2 + (y - 0.5)^2$ and can be computed as (e.g., for the first fish) 0.149926 = $(0.663165 - 0.5)^2 - (0.851145 - 0.5)^2$. The next column is "pbest" (or personal best) shaded in blue . On day-0, given no history, these values are just the same values as the objective function. The next two columns record the coordinates the pbest of each fish. The next column is to select the best fish and, in the example, fish 2 is the "gbest" (or global best), as it has the lowest objective function value 0.111287, with the next two columns recording its coordinates. Lastly, in the last two columns, we have the velocity of each fish. On day-0, these are also randomly chosen from uniform distribution.

**Table A1.** A Demonstration of PSO (five particles, two dimensions).

| day 0 | coordinates | | obj fcn | pbest | pbest coordinates | | gbest | | | velocity | | | |
|---|---|---|---|---|---|---|---|---|---|---|---|---|---|
| | x | y | | | x | y | if yes? | x | y | x | y | | |
| fish 1 | 0.663165 | 0.851145 | 0.149926 | 0.149926 | 0.663165 | 0.851145 | 0 | | | 0.657494 | 0.10171 | | |
| fish 2 | 0.280451 | 0.751167 | 0.111287 | 0.111287 | 0.280451 | 0.751167 | 1 | 0.280451 | 0.751167 | 0.267828 | 0.407284 | | |
| fish 3 | 0.406239 | 0.144589 | 0.135108 | 0.135108 | 0.406239 | 0.144589 | 0 | | | 0.140151 | 0.045138 | | |
| fish 4 | 0.976293 | 0.41713 | 0.233723 | 0.233723 | 0.976293 | 0.41713 | 0 | | | 0.954009 | 0.153271 | | |
| fish 5 | 0.898113 | 0.069016 | 0.344241 | 0.344241 | 0.898113 | 0.069016 | 0 | | | 0.622724 | 0.229386 | | |
| | | | | | | | 0.111287 | 0.280451 | 0.751167 | | | | |

| day 1 | coordinates | | obj fcn | pbest | | | gbest | | | velocity | | r1 | r2 |
|---|---|---|---|---|---|---|---|---|---|---|---|---|---|
| | x | y | | | x | y | if yes? | x | y | x | y | | |
| fish 1 | 1.32066 | 0.952855 | 0.87856 | 0.149926 | 0.663165 | 0.851145 | 0 | | | 0.569697 | 0.085779 | 0.093361 | 0.617054 |
| fish 2 | 0.548279 | 1.158451 | 0.435888 | 0.111287 | 0.280451 | 0.751167 | 0 | | | 0.241045 | 0.366555 | 0.41755 | 0.699711 |
| fish 3 | 0.54639 | 0.189727 | 0.098421 | 0.098421 | 0.54639 | 0.189727 | 1 | 0.54639 | 0.189727 | 0.131175 | 0.13099 | 0.159044 | 0.862283 |
| fish 4 | 1.930303 | 0.570401 | 2.050722 | 0.233723 | 0.976293 | 0.41713 | 0 | | | 0.547117 | 0.287474 | 0.509094 | 0.879301 |
| fish 5 | 1.520838 | 0.298403 | 1.082751 | 0.344241 | 0.898113 | 0.069016 | 0 | | | 0.427 | 0.353833 | 0.383035 | 0.564073 |
| | | | | | | | 0.098421 | 0.54639 | 0.189727 | | | | |

| day 2 | coordinates | | obj fcn | pbest | | | gbest | | | velocity | | r1 | r2 |
|---|---|---|---|---|---|---|---|---|---|---|---|---|---|
| | x | y | | | x | y | if yes? | x | y | x | y | | |
| fish 1 | 1.890357 | 1.038635 | 2.22322 | 0.149926 | 0.663165 | 0.851145 | 0 | | | 0.470934 | 0.057302 | 0.038976 | 0.535754 |
| fish 2 | 0.789324 | 1.525006 | 1.134346 | 0.111287 | 0.280451 | 0.751167 | 0 | | | 0.072254 | −0.19386 | 0.538003 | 0.584523 |
| fish 3 | 0.677565 | 0.320717 | 0.063672 | 0.063672 | 0.677565 | 0.320717 | 1 | 0.677565 | 0.320717 | 0.165733 | 0.165499 | 0.363449 | 0.925142 |
| fish 4 | 2.477419 | 0.857875 | 4.038262 | 0.233723 | 0.976293 | 0.41713 | 0 | | | −0.63784 | 0.024366 | 0.701198 | 0.475362 |
| fish 5 | 1.947837 | 0.652236 | 2.119409 | 0.344241 | 0.898113 | 0.069016 | 0 | | | 0.301537 | 0.298678 | 0.065913 | 0.649501 |
| | | | | | | | 0.063672 | 0.677565 | 0.320717 | | | | |

Now, we can proceed to day-1 using equation. The two coordinates of fish 1 on day-1 are computed, as follows:

$$1.320659 = 0.663165 + 0.657494 \text{ and}$$
$$0.952855 = 0.952855 + 0.101710$$

The new coordinates will lead to new values of the objective function and each fish will update its own pbest and the new gbest will be chosen. As in the example, fish 3 has updated its pbest from 0.135108 to 0.098421, while the other fish will remain the same. Now the gbest is fish 3, no longer fish 2. As we can see the coordinates or pbest and gbest are recorded.

The velocity of fish 1 requires two extra random numbers $r_1$ and $r_2$ that are presented in the last two columns ( shaded in green ) of the table. These two random variables are the "intelligence" of the model. Without them, the model is totally deterministic and is not capable of finding the global optimum.

$$0.569697 = 0.9 \times 0.657494 + 0.093361 \times 1 \times (0.663165 - 0.663165 + 0.617054 \times 1 \times (0.280451 - 0.663165))$$
$$0.085779 = 0.9 \times 0.101710 + 0.093361 \times 1 \times (0.851145 - 0.851145 + 0.617054 \times 1 \times (0.751167 - 0.851145))$$

Now the process repeats itself from day-1 to day-2. In less than 20 iterations, PSO will reach the final coordinate, which is 0.5 and 0.5.

*(b) PSO in Option Pricing*

In Chen et al. (2021), PSO is applied to American option pricing in the following manner. Carr (1998) and Carr et al. (2008), among others, argued that the value of an American option can be calculated as:

$$C = \max_{B(\tau)} e^{-r(\tau - t)} \hat{\mathbb{E}}_t[\max\{K - S(\tau), 0\}] \tag{A2}$$

where $t$ is current time; $r$ is the risk-free rate (constant); $K$ is the strike price; $B(\tau)$ is the exercise boundary value at time $\tau$; $S(\tau)$ is the spot price at time $\tau$; $\tau$ is the exercise time (when $S(\tau) \leq B(\tau)$), and $\hat{\mathbb{E}}_t[\cdot]$ is the conditional risk-neutral expectation.

Equation (18) indicates that an American option value is equal to the expected exercise value when the optimal boundary is specified. However, the challenge lies in the difficulties

in finding the function form of the boundary function $B(\tau)$. Chen et al. reviewed the existing literature of various $B(\tau)$ functions (all of which are parametric) and proposed the use of the PSO to identify a non-parametric boundary function.

Here, we use PSO in parallel to Chen et al. but the expected value is computed from the payoff of the swing contract.

## Notes

[1]   According to the New York Times https://www.nytimes.com/2021/02/20/us/texas-storm-electric-bills.html, (accessed on 10 May 2022) "Scott Willoughby, a 63-year-old Army veteran who lives on Social Security payments in a Dallas suburb. He said he had nearly emptied his savings account so that he would be able to pay the $16,752 electric bill charged to his credit card—70 times what he usually pays for all of his utilities combined ... "

[2]   Barrera-Esteve et al. (2006) compared three methods (forest of trees, MLCS, and dynamic programming) and found them to be indistinguishable and interested readers are encouraged to read their survey paper.

[3]   The optimal quantization method is a method coming from signal processing devised to approximate a continuous signal by a discrete one in an optimal way.

[4]   Many swing contracts involve multiple assets (different gas prices at different terminals). Modeling these complex swing contracts require non-linear optimization in the decision making process.

[5]   A quantitative website named hpcquantlib (https://hpcquantlib.wordpress.com/2011/05/29/swing-option-i-linear-vs-dynamic-programming, accessed on 10 May 2022) discusses how a simple linear programming can be used in substitution for dynamic programming.

[6]   For details, please see Merton's intertemporal asset pricing model (Merton 1973).

[7]   It is also worth noting that under certain regularity conditions the optimal consumption strategy can be shown to be of a so-called bang-bang type, i.e., at each exercise date the optimal decision is to either buy the maximum or minimum quantity (see Theorem 3.1 in Barrera-Esteve et al. 2006).

[8]   We assume the readers have certainly familiarity with the Black–Scholes model and understand the two probabilities usually called Nd1 and Nd2.

[9]   Similar to PSO, an ACO (ant colony optimization) by Dorigo et al. (2000) and ACS (ant colony system) by Dorigo and Gambardella (1997) are both based upon swarm intelligence. The first ant system is first developed by Dorigo and Gambardella (1997) and then popularized by Dorigo et al. (2000).

[10]   The reason is that, as a particle is approaching the global best, the velocity should approach 0 (i.e., the particle should no longer move at the global optimum).

[11]   We implemented a reinforcement learning model on the limited case. The result is available upon request.

[12]   In the "Selecting a Solving Method", choose "Simplex LP".

[13]   Note that with different random numbers the result will change. Yet the convergence is equally fast for this simple problem.

[14]   https://www.cmegroup.com/trading/energy/natural-gas/natural-gas_quotes_globex.html (accessed on 10 May 2022).

[15]   https://www.eia.gov/dnav/ng/hist/rngwhhdm.htm (accessed on 10 May 2022).

[16]   The CPU time is 13 s on an ASUS (LAPTOP-4MUP54NS) Intel i7-8550U CPU @ 1.80GHz 1.99 GHz and 16.0 GB memory.

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
