# Peer review of "A New Look at the Swing Contract: From Linear Programming to Particle Swarm Optimization"

_jrfm, doi:10.3390/jrfm15060246_

Round 1

Reviewer 1 Report

In the present paper, the authors study two kind of things related to Swing contract. First, with a slight modification to the problem, the price can be closely approximated using linear programming which is very fast. Secondly, an artificial intelligent method – particle swarm optimization (PSO), can be used to replace the complex lattice or PDE method.

The problem of study is good but not very novel. Terminologies used in the paper are standard. But the authors have written paper in very rough style. This paper needs major improvements. The following are suggestions and comments:

  • Writing style of paper must be improve.
  • Avoid the footnotes in paper. Write every detail inside paper as definition, remarks, notes etc.
  • In Abstract: It should be rewritten. It should write clearly, what has been done in this paper?
  • Page 10, lines 6-7 from bottom: Needs correction in sentence“…quantities random or not does not impact the solution …”.
  • Page 12 and Page 13: Equation (9) and Equation (13), both are same (repeated). Write equation one place only.
  • Page 14, line 6 from top: Needs correction in sentence“…demonstrate a is limited model …”.
  • References: Mostly references are little old. Add some recent references from the last 5 years and reduce some old references. Based on recent references, enhance introduction.

Note: The highlight all changes by color in revision.

Author Response

We completely rewrote the Abstract, Section 2 (The Swing Contract and the History of Pricing Models) which contains a literature review and our contributions, and Conclusion to reflect the suggestions by the referee. We also improved the contents of the modeling and demonstration sections (Sections 3 and 4) to make it more clear (e.g. we added a comparison discussion to the existing ML models and commented on why the industry can use simpler LP or DP models with a futures price curve to approximate the swing option price and identifies situations where such approximations shall fail).

We thank the referee for his/er suggestions.   Point-by-point responses are given below.

  1. Writing style of paper must be improve.

Modified as suggested.   In particular, Section 2 of the paper “The Swing Contract and the History of Pricing Models” was completely rewritten to better classify the long and complex literature (in the swing contract valuation). Abstract and Conclusion are also completely rewritten to better reflect the motivation and contributions of the paper.

  1. Avoid the footnotes in paper. Write every detail inside paper as definition, remarks, notes etc.

These footnotes were meant to provide auxiliary information and not directly related to the paper. But given the suggestion, we have moved as much as we can those footnotes (especially long ones) to the paper. Those that remain are truly suitable (e.g. a website URL) to stay as footnotes. Those that have been moved to the text (after certain modifications) are #3, 4, 6 ~ 8, 10, 11, 15, 18 and 19.

  1. In Abstract: It should be rewritten. It should write clearly, what has been done in this paper?

The Abstract was completely rewritten. It is shorter and better related to the motivation of the paper.

  1. Page 10, lines 6-7 from bottom: Needs correction in sentence“…quantities random or not does not impact the solution …”.

Corrected – “quantities random or not does not impact the solution”. Other spelling and grammatical errors are also corrected.   The paper has been run through Google Doc for spelling and grammatical errors.

  1. Page 12 and Page 13: Equation (9) and Equation (13), both are same (repeated). Write equation one place only.

Corrected – [equation (13) removed.]

  1. Page 14, line 6 from top: Needs correction in sentence“…demonstrate a is limited model …”.

Corrected “demonstrate a is limited model...” Other spelling and grammatical errors are also corrected.

  1. References: Mostly references are little old. Add some recent references from the last 5 years and reduce some old references. Based on recent references, enhance introduction.

More recent (mainly machine learning) papers are included. Note that there is a gap (between “old” and “new”) in the literature due to market activities and change in methodologies (from traditional option pricing to AI/ML).

Reviewer 2 Report

There are following some suggestions/comments

  1. There are many grammatical, typographical errors and inconsistent mathematical expressions in the paper (text and mathematical expressions are not matched). They can be easily seen from the text. Read carefully.
  2. What is your main contribution? The contribution of the current work should be emphasized in the introduction. Mention the contribution in terms of continuities, curvature properties etc. Give reasons in details.
  3. What kind of reason sent you to study this topic?
  4. What are the limitations and benefits of your work?
  5. Add CPU time for computations for some computational values.
  6. The abstract, conclusion and introduction need to be re-written/revised properly, in terms of the suggestions/professional way (avoid using we, our, etc…).
  7. All figures are needed in more attractive way.
  8. Some abbreviations are not clear.
  9. The abstract is very long try to reduce and comprehensive.

The article is needed to organize properly. The viewpoint in this article and the results have a bit on merit and are of little interest, thus may it reasonable to consider the publication of this paper in “JRFM”

I recommend this paper for publication subject to the above minor changes aimed at improving the quality of the article.

Author Response

We completely rewrote the Abstract, Section 2 (The Swing Contract and the History of Pricing Models) which contains a literature review and our contributions, and Conclusion to reflect the suggestions by the referee. We also improved the contents of the modeling and demonstration sections (Sections 3 and 4) to make it more clear (e.g. we added a comparison discussion to the existing ML models and commented on why the industry can use simpler LP or DP models with a futures price curve to approximate the swing option price and identifies situations where such approximations shall fail).

We thank the referee for his/er suggestions.   Point-by-point responses are given below.

  1. There are many grammatical, typographical errors and inconsistent mathematical expressions in the paper (text and mathematical expressions are not matched). They can be easily seen from the text. Read carefully.

The entire text has been run through Word Doc and Google Doc to check for spelling and grammatical errors.

  1. What is your main contribution? The contribution of the current work should be emphasized in the introduction. Mention the contribution in terms of continuities, curvature properties etc. Give reasons in details.

The motivation and contributions are now clearly stated in the Abstract, Section 2 (The Swing Contract and the History of Pricing Models), and Conclusion (which have been completely rewritten).

  1. What kind of reason sent you to study this topic?

It is two-fold: recent interest (and soaring trading volume) in the energy derivative market and the use of AI/ML in quantity valuation – which are now clearly stated in the Abstract and Conclusion.

  1. What are the limitations and benefits of your work?

This is explained in the newly rewritten Conclusion.

  1. Add CPU time for computations for some computational values.

This is only possible for the PSO calculations, because the others (e.g. LP) are done via Excel spreadsheets (via Solver).   Hence, we modified footnote #23 (now footnote #16) to reflect the computation time.

  1. The abstract, conclusion and introduction need to be re-written/revised properly, in terms of the suggestions/professional way (avoid using we, our, etc…).

It is more customary now in journal articles to write in first person. To satisfy the suggestion here, we modified as much as we can to avoid colloquial type of expressions (a number of changes were made as a result).   Abstract, Conclusion and Section 2 of the paper “The Swing Contract and the History of Pricing Models” where motivation and major contributions are discussed have been completely rewritten.

  1. All figures are needed in more attractive way.

After going through the Tables, the only one that may cause some confusion is Table 1 which explains the detailed mechanics of PSO.   Admittedly, this is more of an appendix material than a table in the main text. Hence, we moved it to the Appendix accordingly. We also added some more explanatory texts to Tables 7 and 8 to fulfill the suggestion here.

  1. Some abbreviations are not clear.

We went through all the abbreviations of the paper and made some corrections as suggested.

  1. The abstract is very long try to reduce and comprehensive.

The Abstract was completely rewritten. It is shorter (as requested) and better related to the motivation of the paper, which is also to satisfy comment #3 above.

Round 2

Reviewer 1 Report

The authors have included all my suggestions and comments in the paper. We recommend paper for publication in JRFM.